# Current Sodium Intakes in the United States and the Modeled Effects of Glutamate Incorporation into Select Savory Products

**DOI:** 10.3390/nu11112691

**Published:** 2019-11-07

**Authors:** Taylor C. Wallace, Alexandra E. Cowan, Regan L. Bailey

**Affiliations:** 1Department of Nutrition and Food Studies, George Mason University, Fairfax, VA 22030, USA; taylor.wallace@me.com; 2Think Healthy Group, Inc., Washington, DC 20036, USA; 3Department of Nutrition Science, Purdue University, West Lafayette, IN 47907, USA; acowan9@purdue.edu

**Keywords:** sodium, glutamate, NHANES, usual intake, chronic disease risk reduction

## Abstract

Most Americans have dietary sodium intakes that far exceed recommendations. Given the association of high sodium with hypertension, strategies to reduce sodium intakes are an important public health target. Glutamates, such as monosodium glutamate, represent a potential strategy to reduce overall intakes while preserving product palatability; therefore, this project aimed to model sodium replacement with glutamates. The National Cancer Institute method was used to estimate current sodium intakes, and intakes resulting from glutamate substitution (25%–45%) in a limited set of food groups for which substitution is possible. Data sets for individuals aged ≥1 year enrolled in the U.S. National Health and Nutrition Examination Survey 2013–2016 (*n* = 16,183) were used in the analyses. Glutamate substitution in accordance with the U.S. Department of Agriculture’s food codes was modeled by conservatively altering estimates of sodium intake reductions derived from the published, peer-reviewed literature. The addition of glutamates to certain food categories has the potential to reduce the population’s sodium intake by approximately 3% overall and by 7%–8% among consumers of ≥1 product category in which glutamates were substituted for sodium chloride. Although using glutamates to substitute the amount of sodium among certain food groups may show modest effects on intakes across the population, it is likely to have a more substantial effect on individuals who consume specific products.

## 1. Introduction

Dietary sodium reduction is an important goal for the improvement of public health, as reduced sodium intake has been shown to decrease hypertension risk [1]. Hypertension is a valid surrogate endpoint reflective of risks for a myriad of cardiovascular diseases, a leading source of mortality in U.S. adults [2]. Many multifaceted policy and education initiatives aimed at reducing sodium intakes have been ongoing for decades. It has been estimated that a 40% reduction in the U.S. population’s intake of sodium over 10 years may save at least 280,000 lives [3] and drastically reduce the number of disability-adjusted life years (DALYs). Despite ongoing public health education and policy initiatives, the preponderance of the U.S. population exceeds current recommendations for sodium intake [4]. Among hypertensive adults, 86% exceed 2300 mg dietary sodium/day [5]. 

Trends in sodium intakes have not changed over the past 10 years (five U.S. National Health and Nutrition Examination Survey (NHANES) cycles) [6]. Those with the lowest household education, non-Hispanic black race/Hispanic origin, and lowest income have seen the largest increase in sodium intakes derived from snack foods from 1977–1978 to 2011–2014 [7]. Sodium is primarily consumed as sodium chloride (NaCl) and the majority of intake is derived from sodium added during food processing for the purposes of providing flavor or food safety properties. NaCl may also be used as a processing aide. More than 70% of sodium intake in the United States comes from commercially processed and restaurant foods and no single category comprises >7% of total intake [8]. With the exception of milk, which naturally contains sodium, the top ten food categories contributing sodium to the diet of U.S. school-aged children are composed of foods in which sodium is added during processing or preparation [9]. Yeast breads contribute the most sodium to the U.S. diet (for individuals aged ≥2 years) [8].

Sodium reduction poses technical challenges given its role in the palatability and safety of food (e.g., preventing bacterial growth and spoilage) [10]. When salt is reduced, palatability and consumer acceptance of a product generally tends to decrease. In 2016, the U.S. Food and Drug Administration (FDA) published draft guidance on voluntary sodium reduction goals for public comment with an aim to reduce U.S. daily intake from 3400 mg to 3000 mg within 2 years (short-term goal) and to 2300 mg within 10 years (long-term goal) [11]. This gradual process set reduction targets for >150 food categories. 

Currently, no perfectly viable alternative for replacing sodium exists in the contemporary food marketplace, although several innovations do exist among various product categories. For example, glutamate, a nonessential amino acid, has been used to enhance the taste and palatability of food. Indeed, the 2010 Institute of Medicine (now the Institute of Medicine) report, *Strategies to Reduce Sodium Intake in the United States*, emphasizes that achieving lower intakes of excessive sodium should be a critical focus for all Americans and it is possible to use monosodium glutamate (MSG), the most common glutamate salt and flavor enhancer, to lower the overall sodium level in certain foods while maintaining palatability [10]. MSG contains about 12% sodium, which is less than one-third of that contained in table salt (39%) [12]. There is an appropriate amount of MSG that can be used to replace salt while maintaining consumer acceptance in food [12]. Excess MSG does not promote umami taste, and to the contrary, may negatively impact the taste profile of food, most notably at levels >1% [13]. Although MSG is the most widely used flavor enhancer in food, other effective glutamate salts, such as calcium di-glutamate, exist but do not provide as pronounced of an effect. A considerable number of studies have demonstrated that various forms of glutamate can help reduce the amount of sodium in specific foods, including soups, prepared dishes, processed meat, and dairy products, by enhancing palatability [14,15,16,17,18,19,20,21]. However, much less is known about how glutamate substitution would affect sodium intakes at the population level across a range of different foods. Therefore, the purpose of this study was to first estimate contemporary sodium intakes of the U.S. population, and then to model MSG substitution in select products—with substantial supportive literature to ensure feasibility—to estimate potential population-level reductions in sodium intakes. 

## 2. Materials and Methods 

### 2.1. Study Population

The NHANES, conducted by the Centers for Disease Control and Prevention (CDC) National Center for Health Statistics (NCHS), is a nationally representative, continuous, cross-sectional survey of noninstitutionalized, civilian residents of the United States [22]. Since 1999, the NHANES protocol has included an in-person household interview component and a follow-up health examination in the mobile examination center (MEC) for each participant. The NHANES survey protocol was approved by the CDC NCHS Research Ethics Review Board, and written informed consent was obtained for all survey participants or proxies [22]. Data from NHANES 2013–2016 were combined for these analyses. Pregnant and/or lactating women (*n* = 112) were excluded, yielding a combined sample of 16,183 participants who had completed and provided 24-hour dietary intake data. Further analyses evaluating the contributions of select food categories to total sodium intake from the diet were limited to U.S. adults (≥19 years), excluding those aged ≤19 years (*n* = 6071), yielding a final analytic sample size of 10,112 U.S. adults (≥19 years).

### 2.2. Demographic Data

All demographic data used for this analysis, including data on sex and age, were collected from participants using the computer-assisted personal interview system during the household interview. Age was categorized to be consistent with the dietary reference intake (DRI) age groups, defined as 1–3, 4–8, 9–13, 14–18, 19–30, 31–50, 51–70, and ≥71 years, and was used to compare estimates of sodium intakes. Children and adults were defined as those individuals who were aged 1–18 and ≥19 years, respectively.

### 2.3. Dietary Sodium Intake

NHANES participants were asked to complete two 24-hour dietary recalls for the collection of dietary intake data. The first 24-hour dietary recall was self-reported in the MEC and collected in person by trained NHANES interviewers. The second 24-hour dietary recall interview was completed via telephone approximately 3–10 days after the MEC examination. Both 24-hour recalls were collected by trained interviewers using the U.S. Department of Agriculture’s (USDA) validated, automated, multiple-pass method [23,24]. Proxy respondents provided dietary intake data for young children and proxy-assisted interviews were utilized for children aged 6–11 years. Questionnaires, data sets, and all related documentation from each NHANES cycle can be found on the NCHS website [25]. The USDA Food and Nutrient Database for Dietary Studies was used to convert foods and beverages (as reported) to their respective sodium intake values [26].

### 2.4. Comparison to DRI Values

The DRIs are a set of nutrition reference values, defined by the National Academies of Sciences, Engineering and Medicine (NASEM) Food and Nutrition Board, that are designed to assess nutrient intakes of healthy people and establish guidelines for risk assessment in the United States and Canada [27]. The DRIs for sodium and potassium were recently updated in 2019, and for the first time, a new category of DRIs based on chronic disease, called the chronic disease risk reduction (CDRR), was established for sodium [1]. Other DRIs established by the NASEM for sodium include the estimated average requirement, recommended dietary allowance, adequate intake, and tolerable upper intake level (UL). DRI values differ for individuals based on age and sex [1]. Sodium was reported as usual intake and the proportion of the population with intakes above the CDRR and UL. Information regarding the recent DRIs for sodium and potassium can be found in the NASEM report, *Dietary Reference Intakes for Sodium and Potassium* [1].

### 2.5. Sodium Intake Modeling

Glutamates, such as MSG, are flavor enhancers that have been effectively used to reduce sodium in certain food categories, particularly in savory products. A review of the scientific literature demonstrates that glutamates have been utilized to reduce sodium among various mainstream products (Table 1). Assuming that the food supply already contains a significant amount of glutamates and that amounts used among products vary, we made conservative assumptions, in consultation with food scientists, of a 25%–45% reduction in sodium by substitution of sodium chloride with glutamate salts across certain categories of foods using the USDA food codes (Table 2). Consumers of glutamates were those who reported consumption of one or more food categories in which glutamates were substituted for sodium chloride.

### 2.6. Statistical Analyses

The National Cancer Institute (NCI) method was used to determine estimates of usual intake of sodium from the diet. Covariates used in the NCI model were as follows: (1) sequence of 24-hour recall (first or second dietary recall); and (2) day of the week the 24-hour recall was collected (weekend/weekday). All statistical analyses were performed using SAS software (version 9.3; SAS Institute Inc., Cary, NC, USA). SAS macros necessary to fit this model and to perform estimation of usual intake distributions, as well as additional details and resources concerning the NCI Method, are available via the NCI website [30]. The fitted model is a two-part model that first uses logistic regression to estimate the probability of intake consumption for each consumer, and then, secondly, uses linear regression to estimate the actual daily amount of intake on a transformed scale, while taking into account within-person variation [30]. Sample weights were used to account for differential nonresponse and noncoverage and to adjust for planned oversampling of some groups, in order to generate a nationally representative sample. Standard errors for all statistics of interest were approximated using Fay’s modified, balanced, repeated-replication technique [31,32].

## 3. Results

### 3.1. Current Mean Sodium Intakes, Percentages above the CDRR, and Percentages above the UL

Overall, sodium intakes among the general U.S. population are higher than federal recommendations. On the basis of NHANES 2013–2016 data, Americans (aged >1 year) consume approximately 3361 mg sodium/day on average (Table 3). Mean daily sodium intake from foods and beverages among the U.S. population was 2906 mg/day for children (aged 1–18 years) and 3499 mg/day for adults (aged ≥19 years). 

Regardless of age, men had higher sodium intakes than women. Specifically, among adults (≥19 years), men typically consumed approximately 4067 mg sodium/day, whereas women only consumed approximately 2956 mg sodium/day. Similar themes were apparent among children (1–18 years); boys had higher mean sodium intakes than girls (3268 versus 2673 mg/day, respectively). Therefore, women of all ages were less likely to exceed the CDRR and UL compared to men.

Across age subgroups, sodium intake was highest among men and women aged 19–30 years (4431 versus 3138 mg/day, respectively) and varied across the life course. For men, sodium intake increased with age in the adolescent years (1–18 years), plateaued among early adulthood (19–30 years), and then decreased through the remainder of adulthood (≥31 years). However, slightly different patterns were observed among women. Whereas sodium intakes among young girls increased with age until 9–13 years, a slight decrease in intake was observed between the ages of 14–18 years, followed by an increase in intake from 19–30 years, and lastly, a final decrease in intake for the remainder of adulthood (≥31 years). Older adults (≥71 years) had the lowest sodium intakes of all adult participants among both men and women. Thus, younger adults were more likely to exceed the CDRR and UL for sodium compared to their older adult (≥71 years) counterparts. Among children, boys and girls in the 4–8-year and 9–13-year life stages had the highest prevalence of exceeding the CDRR and UL. 

Estimated mean sodium intake and the percent-wise contributions for selected food categories to total sodium intake in the diets of U.S adults (≥19 years) and children (1–18 years) are presented in Table 4 and Table 5. On a population level, no individual food group contributes large amounts of sodium to the diet; sodium intakes appear to be widespread throughout the food supply. However, the savory food groups represented in Table 4 and Table 5 provide much larger proportions of sodium to the diets of those who consumed these products (i.e., “consumers”). For example, meat-based frozen meals provide 0.3% of the sodium present in the diets of all U.S adults, but as much as one-third (32%) among consumers. Among children, the top three contributors to total sodium intakes are crackers and salty snacks, cured meats, and select cheeses (Table 5). Although intakes of these select food categories remain high overall, intakes among consumers of these categories are significantly higher than the general population of U.S. children (Figure 1). For cured meats in particular, children who consume these products receive 20% of their usual sodium intake from this source, whereas cured meats account for only 4% of total sodium intakes among the general population of U.S. children. 

### 3.2. Models of the Effects of Gutamates on Mean Usual Sodium Intakes, Percentages above the CDRR, and Percentages above the UL

Universal incorporation of glutamates into the select savory food groups (presented in Table 2) would result in a 3% (162 mg/day) reduction in overall sodium intakes in the U.S. population (aged ≥ 1 year) and a 7–8 percentage point reduction among consumers of one or more food categories in which sodium chloride could be substituted for by glutamates. Among U.S. children specifically, glutamates have the potential to reduce the proportion of the population exceeding the UL for sodium by 5 percentage points (Table 3) and to reduce consumer intakes by 211–263 mg/day among boys and girls. Likewise, glutamates could reduce the sodium intakes of consumers by 321 and 236 mg/day in adult men and women, respectively (Table 6 and Table 7). 

## 4. Discussion

Dietary factors are among the top contributors to chronic disease risk and DALYs in the United States. High intake of sodium is one risk factor that has been identified as a contributor to this burden [33]. According to our analysis, current estimates of mean usual sodium intake remain high across all age and sex subgroups of the U.S. population, and they continue to exceed authoritative recommendations, consistent with previous reports [4,6]. Contrary to prior reports [6], age-related differences in estimated usual intakes did not appear to be more pronounced in men than women (Table 3). Our analyses provide unique data on the diets of consumers of select food categories. Although reducing the amount of sodium among certain food groups may show modest effects on intakes across the population, it may have a large effect on individuals who consume these types of products. For example, about 18.7% of U.S. adults consume cured meats on any given day; reducing sodium intake of cured meat products by 40% would have a large impact on those consumers, since cured meats account for 21% of their total sodium intake from the diet. Figure 1 illustrates substantial effects of incorporating glutamates into crackers and salty snacks, cured meats, and select cheeses on U.S. children (1–18 years). Given current sources of sodium in the diets of young children, glutamate’s use in these select categories may have the greatest impact on sodium intakes in this age group. Meat-based frozen meals, vegetable and meat-based soups, and cured meats contain the highest amounts of sodium; however, grain products were previously shown to be the largest contributor of sodium intakes to the U.S. diet because they are more ubiquitously consumed [6]. Consumer sentiment around MSG has deterred many consumer-packaged food companies from utilizing it to reduce sodium intake in products [34]. 

Several other strategies have been applied to reduce the population’s sodium intake. Gradual reduction through a cumulative series of small decreases over 6 weeks was shown to be effective in reducing the sodium content of white bread by about 25% without altering palatability. However, reduction would need to be applied to all breads on the market to go unnoticed by consumers [35]. Potassium chloride (KCl), calcium chloride (CaCl_2_), and magnesium sulfate (MgSO_4_) have been used as substitutes for table salt; however, their bitter taste has limited their use and uptake by consumers [36]. Citric acid in tomato soup [37] and lactic acid in bread [38] have the potential to enhance saltiness and be useful in reducing sodium intake. SODA-LO, a new but more costly sodium-reduction ingredient that can reduce sodium in certain applications through its technology that turns standard salt crystals into free-flowing, hollow salt microspheres, has been shown to deliver taste and function by maximizing surface area in products such as potato chips and baked goods [6]. 

Current intake above public health recommendations is not solely a United States-centric issue, but a global pandemic demonstrated by high intakes of sodium in other countries [39]. Assuming the accuracy of NHANES and other international databases, ongoing public health education initiatives show no signs of success in decreasing intakes. Frequent use of nutrition labels appears to be associated with lower consumption of sodium and high-sodium foods; however, while surveys suggest that consumers may wish to reduce their sodium intake, it is likely not a priority in what most consumers choose to eat [40]. In fact, evidence indicates that many consumers avoid products labeled as “low sodium” [40]. Reducing intakes through food science and technological advances seems appropriate, in order to make the most impactful reductions in the consumption of sodium at the population level. 

Our study has some strengths and limitations to consider. First, the strengths of our analysis are that the models applied to examine usual intakes adjusted for the effects of within-person variation measurement error, and that NHANES is a large nationally representative sample that allows for the estimation of usual intakes of sodium at the population level. However, the limitations of our study should also be noted. This modeling study used conservative assumptions of sodium reduction by substituting glutamates for sodium chloride in several What We Eat in America food categories. We chose these conservative reduction values, presented in Table 2, upon consultation with food scientists, with the hopes of accounting for what is already contained in the food supply so as not to overestimate the total effect of glutamates. Restaurant foods supply a large portion of sodium to the U.S. diet [40]; however, we chose to not model inclusion of glutamates into restaurant foods, since many of these flavor enhancers are already in widespread use in restaurants. Therefore, the effect of glutamates could be greater than what is presented in our study. NHANES also has several limitations, including the reliance on self-reported dietary intake data and assumptions of USDA reference database accuracy for estimating sodium intakes across the population. Self-reported dietary data are prone to systematic errors, such as energy underreporting. Additionally, we cannot completely rule out the potential for self-selection bias; that is, people who participate in nutrition- and health-related research tend to differ by sociodemographic factors and may have been more interested in participating in NHANES [41]. Finally, in order to fully maximize the effectiveness of sodium reduction, the acceptability of MSG among consumers must be taken into consideration [42,43].

## 5. Conclusions

Current sodium intakes in the United States remain high and unchanged from previous NHANES cycles, exceeding public health recommendations. The addition of glutamates to certain savory food categories has the potential to help reduce the population’s intake of sodium by approximately 3.0%, and to reduce the intake by 7.3% among consumers of the product categories in which sodium chloride could be substituted for by glutamates. While reducing the amount of sodium among certain food groups may show modest effects on intakes across the adult population, it may have a large effect on those who consume those types of products. 

## Figures and Tables

**Figure 1 nutrients-11-02691-f001:**
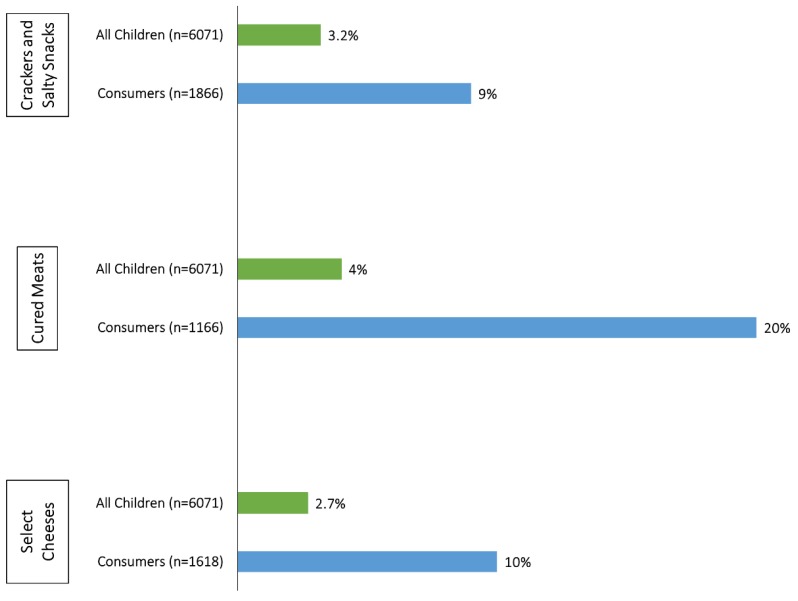
Percentage-wise contributions from selected food groups on total sodium intake (in milligrams) in the diets of U.S. children (aged 1–18 years), National Health and Nutrition Examination Survey 2013–2016. Based on FDA-NIH Biomarker Working Group [2] and U.S. National Academies of Sciences [1].

**Table 1 nutrients-11-02691-t001:** Sodium reduction in various food products with the incorporation of glutamates.

Reference	Food	Sodium Reduction (%)
Ball et al. [14]	Soups	40
Yamaguchi and Takahashi [15]		
Roininen et al. [16]		
Jinap et al. [17]	Spicy soups	33
Carter et al. [18]	Chicken broth	38
dos Santos et al. [19]	Sausage	50
Rodrigues et al. [28]	Mozzarella cheese	54
de Quadros et al. [20]	Fish burgers	50
Buechler [29]	Chips and rice puffs	51
Leong et al. [21]	Chicken rice mixed dish	31
Leong et al. [21]	Mee soto broth	22

**Table 2 nutrients-11-02691-t002:** Sodium reduction assumptions due to incorporation of glutamates by USDA food code ^1^

USDA Food Code	Sodium Reduction (%)
1 Milk and Milk Products	
*14 Cheeses*	
140 Cheese, NS as to type	45
141 Natural cheeses	45
144 Processed cheeses and cheese spreads	45
145 Imitation cheeses	45
147 Cheese soups	45
2 Meat, Poultry, Fish, and Mixtures	
*25 Organ meats, sausages and lunchmeats, and meat spreads*	
252 Frankfurters, sausages, lunchmeats, and meat spreads	
2521 Frankfurters	40
2522 Sausages	40
2523 Luncheon meats (loaf)	40
2524 Potted meat, spreads	40
*28 Frozen and shelf-stable plate meals, soups, and gravies with meat, poultry, fish base; gelatin and gelatin-based drinks*	
281 Frozen or shelf-stable plate meals with meat, poultry, fish as major ingredient	
2811 Beef frozen or shelf-stable meals	25
2812 Pork or ham frozen or shelf-stable meals	25
2813 Veal frozen or shelf-stable meals	25
2814 Poultry frozen or shelf-stable meals	25
2815 Fish, shellfish frozen meals	25
2816 Miscellaneous meat frozen or shelf-stable meals	25
283 Soups, broths, extracts from meat, poultry, fish base	
2831 Beef soups	30
2832 Pork soups	30
2833 Lamb soups	30
2834 Poultry soups	30
28345 Poultry cream soups	30
2835 Fish, shellfish soups	30
2836 Puerto Rican soups	30
285 Gravies from meat, poultry, fish base	30
5 Grain Products	
*54 Crackers and salty snacks from grain products*	
540 Crackers, NS as to type	40
543 Nonsweet crackers	40
544 Salty snacks from grain products	40
7 Vegetables	
*71 White potatoes and Puerto Rican starchy vegetables*	
718 Potato soups	30
*72 Dark-green vegetables*	
723 Dark-green vegetable soups	30
*73 Deep-yellow vegetables*	
735 Deep-yellow vegetable soups	30
*74 Tomato and tomato mixtures*	
746 Tomato soups	30
*75 Other vegetables*	
756 Vegetable soups	30
*77 Vegetables with meat, poultry fish*	
775 Puerto Rican stews or soups with starchy vegetables (viandas)	30

^1^ USDA, U.S. Department of Agriculture; NS, not specific.

**Table 3 nutrients-11-02691-t003:** Estimated and potential means usual sodium intake (in milligrams) from dietary sources and the estimated percentages of mean usual-sodium intakes greater than the CDRR and UL in the U.S. population (aged ≥1 year) by age and sex, NHANES 2013–2016 ^1,2^

Age (Years)	*n*	Current Intake			Potential Intake		
		Mean (SE)	% >CDRR (SE)	% >UL (SE)	Mean (SE)	% >CDRR (SE)	% >UL (SE)
All (≥1)	16,183	3360.7 (19.9)	86 (0.4)	85 (0.5)	3198.7 (18.2)	83 (0.5)	82 (0.5)
Children (1–18)	6071	2905.8 (26.9)	88 (0.7)	81 (0.9)	2743.0 (23.4)	85 (0.7)	76 (0.9)
Adults (≥19)	10,112	3499.0 (23.9)	88 (0.6)	88 (0.6)	3330.7 (21.9)	85 (0.6)	85 (0.6)
Men							
1–18	3077	3268.5 (46.2)	94 (0.6)	91 (0.9)	3089.5 (43.7)	92 (0.7)	87 (1.0)
1–3	577	2062.5 (50.6)	95 (1.9)	83 (3.5)	1920.7 (45.1)	92 (2.3)	76 (3.8)
4–8	867	2829.8 (74.6)	100 (0.3)	96 (1.7)	2630.5 (60.1)	99 (0.5)	93 (2.4)
9–13	843	3383.0 (94.1)	100 (0.2)	99 (1.8)	3211.2 (88.6)	100 (0.3)	99 (2.4)
14–18	790	3935.4 (90.8)	94 (1.7)	94 (1.7)	3776.2 (89.8)	93 (1.9)	93 (1.9)
≥19	4955	4066.7 (38.7)	97 (0.5)	97 (0.5)	3872.0 (35.7)	96 (0.5)	96 (0.5)
19–30	1029	4431.8 (86.1)	98 (0.7)	98 (0.7)	4239.0 (80.7)	97 (0.8)	97 (0.8)
31–50	1622	4217.5 (70.2)	98 (0.7)	98 (0.7)	4023.4 (64.6)	98 (0.9)	98 (0.9)
51–70	1606	3878.0 (65.6)	96 (1.0)	96 (1.0)	3678.4 (63.3)	94 (1.2)	94 (1.2)
≥71	698	3357.2 (82.2)	89 (2.4)	89 (2.4)	3174.2 (78.6)	85 (3.0)	85 (3.0)
Women							
1–18	2994	2673.3 (42.0)	86 (1.4)	77 (1.9)	2519.6 (35.6)	81 (1.3)	71 (1.8)
1–3	548	1867.4 (56.9)	90 (3.2)	73 (4.5)	1706.0 (45.8)	84 (4.0)	63 (4.2)
4–8	818	2480.5 (45.7)	99 (0.9)	88 (3.0)	2342.3 (45.2)	97 (1.4)	80 (3.6)
9–13	815	3051.6 (59.3)	98 (1.0)	90 (2.4)	2880.7 (53.8)	97 (1.2)	87 (2.8)
14–18	813	2818.6 (78.8)	73 (4.8)	73 (4.8)	2684.2 (69.5)	67 (4.6)	67 (4.6)
≥19	5157	2955.8 (24.3)	78 (1.1)	78 (1.1)	2812.8 (22.4)	73 (1.1)	73 (1.1)
19–30	977	3138.0 (57.8)	85 (2.9)	85 (2.9)	2995.4 (57.7)	81 (3.2)	81 (3.2)
31–50	1748	3091.1 (55.2)	81 (1.9)	81 (1.9)	2951.8 (53.2)	77 (1.9)	77 (1.9)
51–70	1708	2896.1 (38.8)	77 (2.2)	77 (2.3)	2740.1 (34.7)	71 (2.2)	71 (2.2)
≥71	724	2510.7 (73.4)	59 (4.7)	59 (4.7)	2393.8 (71.3)	52 (4.6)	52 (4.6)

^1^ CDRR, chronic disease risk reduction; UL, tolerable upper intake level; NHANES, National Health and Nutrition Examination Survey; SE, standard error. ^2^ The analytic sample includes individuals aged ≥1 year who were not pregnant or lactating and had complete information for age and 24-hour dietary recall on day one.

**Table 4 nutrients-11-02691-t004:** Estimated mean sodium intake (in milligrams) and percentage-wise contributions from selected food groups in the diets of U.S. adults (aged ≥19 years) by age and sex, NHANES 2013–2016 ^1,2,3^.

Figure	All Adults		Men		Women	
	Consumers	All Adults	Consumers	All Men	Consumers	All Women
Select cheeses						
*n*	3115	10,112	1515	4955	1600	5157
Mean (SE)	303 (8.2)	109 (4.2)	348 (12.6)	127 (6.0)	258 (8.6)	92 (4.4)
% contribution (SE)	8 (0.2)	3 (0.1)	8 (0.3)	3 (0.1)	9 (0.3)	3 (0.1)
Cured meats						
*n*	1886	10,112	1049	4955	837	5157
Mean (SE)	825 (21.0)	159 (6.1)	908 (33.8)	202 (10.1)	715 (23.6)	117 (7.1)
% contribution (SE)	21 (0.4)	4 (0.2)	20 (0.5)	5 (0.2)	22 (0.7)	4 (0.2)
Meat-based frozen meals						
*n **	81	10,112	34	4955	47	5157
Mean (SE)	959 (77.3)	10 (1.5)	1124 (123.1)	7 (1.4)	885 (84.7)	12 (2.5)
% contribution (SE)	32 (2.3)	0.3 (0.0)	33 (4.5)	0.2 (0.0)	31 (2.3)	0.4 (0.1)
Meat-based soups						
*n*	453	10,112	200	4955	253	5157
Mean (SE)	1213 (67.0)	43 (4.7)	1328 (79.3)	40 (5.7)	1129 (77.1)	45 (6.7)
% contribution (SE)	30 (1.2)	1.0 (0.1)	29 (1.7)	0.9 (0.1)	31 (1.6)	1.2 (0.1)
Meat-based gravies						
*n*	255	10,112	133	4955	122	5157
Mean (SE)	441 (41.9)	11 (1.7)	533 (73.2)	14 (3.2)	335 (26.1)	8 (0.8)
% contribution (SE)	11 (0.8)	0.3 (0.0)	11 (1.4)	0.3 (0.1)	10 (0.8)	0.2 (0.0)
Crackers and salty snacks						
*n*	1881	10,112	848	4955	1033	5157
Mean (SE)	267 (9.2)	56 (3.1)	289 (82)	55 (3.0)	250 (14.6)	56 (4.5)
% contribution (SE)	8 (0.3)	1.7 (0.1)	8 (0.3)	1.5 (0.1)	8 (0.4)	1.9 (0.1)
Vegetable soups						
*n*	382	10,112	166	4955	216	5157
Mean (SE)	1003 (54.2)	39 (4.3)	1241 (108.7)	39 (6.2)	848 (58.4)	39 (4.9)
% contribution (SE)	27 (1.3)	1.0 (0.1)	28 (1.9)	0.9 (0.1)	26 (1.7)	1.2 (0.1)

^1^ NHANES, National Health and Nutrition Examination Survey; SE, standard error. ^2^ The analytic sample includes individuals age ≥ 19 years who were not pregnant or lactating and had complete information for age and 24-hour dietary recall on day one. ^3^ Unlike the National Cancer Institute-adjusted statistics that represent long-term, usual dietary intake estimates presented in Table 3, the food intake statistics here are estimates of intake on any given day. The above sample sizes for consumers reflect a categorization of the NHANES respondents based on whether they reported sodium intakes in one of the above categories, and describe the sample, not the population. Estimates reflective of population intakes for the above categories are indicated by “all adults,” “all men,” and “all women,” respectively. * Estimates have SEs of concern or sample sizes of concern.

**Table 5 nutrients-11-02691-t005:** Estimated mean sodium intake (in milligrams) and percentage-wise contributions from selected food groups in the diets of U.S. children (aged 1–18 years) by age and sex, NHANES 2013–2016 ^1,2,3^.

Figure	All Children		Boys		Girls	
	Consumers	All Children	Consumers	All Boys	Consumers	All Girls
Select cheeses						
*n*	1866	6071	921	3077	945	2994
Mean (SE)	277 (7.6)	94 (5.0)	297 (11.5)	101 (6.1)	257 (8.1)	87 (5.5)
% contribution (SE)	9 (0.2)	3.2 (0.1)	9 (0.3)	3.0 (0.2)	10 (0.3)	3.3 (0.2)
Cured meats						
*n*	1166	6071	620	3077	546	2994
Mean (SE)	636 (22.8)	127 (8.2)	677 (28.7)	145 (11.9)	588 (29.7)	109 (8.9)
% contribution (SE)	20 (0.5)	4 (0.3)	20 (0.6)	4 (0.3)	21 (0.8)	4 (0.3)
Meat-based frozen meals						
*n**	14	6071	9	3077	5	2994
Mean (SE)	783 (56.5)	2 (0.7)	671 (61.9)	3 (1.0)	460 (45.8)	1 (1.1)
% contribution (SE)	31 (3.9)	0.08 (0.02)	35 (5.1)	0.1 (0.04)	25 (3.3)	0.05 (0.04)
Meat-based soups						
*n*	175	6071	89	3077	86	2994
Mean (SE)	1097 (73.1)	25 (4.6)	1168 (126.1)	26 (5.5)	1024 (108.3)	23 (6.4)
% contribution (SE)	36 (2.1)	0.8 (0.2)	34 (2.3)	0.7 (0.1)	38 (3.5)	0.9 (0.2)
Meat-based gravies						
*n*	107	6071	62	3077	45	2994
Mean (SE)	405 (56.6)	10 (1.7)	479 (73.2)	14 (2.5)	292 (48.5)	6 (1.6)
% contribution (SE)	10 (1.4)	0.2 (0.04)	11 (1.8)	0.3 (0.1)	9 (1.4)	0.2 (0.05)
Crackers and salty snacks						
*n*	1618	6071	814	3077	804	2994
Mean (SE)	273 (15.3)	75 (6.0)	277 (15.0)	75 (6.5)	268 (23.2)	75 (8.0)
° % contribution (SE)	10 (0.4)	2.7 (0.2)	9 (0.3)	2.5 (0.2)	11 (0.8)	3.0 (0.3)
Vegetable soups						
*n **	111	6071	51	3077	60	2994
Mean (SE)	737 (92.6)	13 (2.7)	812 (145.2)	16 (4.2)	639 (62.5)	10 (2.2)
% contribution (SE)	23 (2.2)	0.4 (0.1)	21 (3.3)	0.4 (0.1)	26 (1.9)	0.4 (0.1)

^1^ NHANES, National Health and Nutrition Examination Survey; SE, standard error. ^2^ The analytic sample includes individuals aged 1–18 years who were not pregnant or lactating and had complete information for age and 24-hour dietary recall on day one. ^3^ Unlike the National Cancer Institute-adjusted statistics that represent long-term, usual dietary intake estimates presented in Table 3, the food intake statistics here are estimates of intake on any given day. The above sample sizes for consumers reflect a categorization of the NHANES respondents based on whether they reported sodium intakes in one of the above categories, and describe the sample, not the population. Estimates reflective of population intakes for the above categories are indicated by “all adults,” “all men,” and “all women,” respectively. * Estimates have SEs of concern or sample sizes of concern.

**Table 6 nutrients-11-02691-t006:** Potential sodium intake reduction with MSG in the diets of U.S. adults (aged ≥19 years) who consume selected food groups by age and select food group, NHANES 2013–2016 ^1,2,3^

Figure	*n*	Potential Reduction, Mean (SE)	Current Intake, Mean (SE)	Potential Intake After Reduction, Mean (SE)	Reduction (%)
Overall sodium (*Reduction among consumers of ≥1 food groups*)					
All consumers	5639	278 (2.7)	3794 (30.1)	3516 (27.4)	7.3 (0.1)
Men	2732	321 (5.1)	4407 (53.0)	4086 (47.9)	7.3 (0.1)
Women	2907	236 (1.2)	3202 (29.1)	2966 (27.9)	7.4 (0.0)
Select cheeses					
All consumers	3115	136 (3.7)	303 (8.2)	167 (4.5)	45.0 (0.0)
Men	1515	156 (5.7)	348 (12.6)	191 (7.0)	45.0 (0.0)
Women	1600	116 (3.9)	258 (8.6)	142 (4.7)	45.0 (0.0)
Cured meats					
All consumers	1886	330 (8.4)	825 (21.0)	495 (12.6)	40.0 (0.0)
Men	1049	363 (13.5)	908 (33.8)	545 (20.3)	40.0 (0.0)
Women	837	286 (9.5)	715 (23.6)	429 (14.2)	40.0 (0.0)
Meat-based frozen meals *					
All consumers	81	240 (19.3)	959 (77.3)	719 (58.0)	25.0 (0.0)
Men	34	281 (30.8)	1124 (123.1)	843 (92.3)	25.0 (0.0)
Women	47	221 (21.2)	885 (84.7)	663 (63.5)	25.0 (0.0)
Meat-based soups					
All consumers	453	364 (20.1)	1213 (67.0)	849 (46.9)	30.0 (0.0)
Men	200	398 (23.8)	1328 (79.3)	930 (55.5)	30.0 (0.0)
Women	253	339 (23.1)	1129 (77.1)	790 (54.0)	30.0 (0.0)
Meat-based gravies					
All consumers	255	132 (12.6)	441 (41.9)	309 (29.4)	30.0 (0.0)
Men	133	160 (22.0)	533 (73.2)	373 (51.3)	30.0 (0.0)
Women	122	101 (7.8)	335 (26.1)	235 (18.3)	30.0 (0.0)
Crackers and salty snacks					
All consumers	1881	107 (3.7)	267 (9.2)	160 (5.5)	40.0 (0.0)
Men	848	115 (3.3)	289 (82)	173 (4.9)	40.0 (0.0)
Women	1033	100 (5.8)	250 (14.6)	150 (8.8)	40.0 (0.0)
Vegetable soups					
All consumers	382	301 (16.3)	1003 (54.2)	702 (37.9)	30.0 (0.0)
Men	166	372 (32.6)	1241 (108.7)	868 (76.1)	30.0 (0.0)
Women	216	254 (17.5)	848 (58.4)	593 (40.9)	30.0 (0.0)

^1^ MSG, monosodium glutamate; NHANES, National Health and Nutrition Examination Survey; SE, standard error. ^2^ The analytic sample included individuals aged ≥ 19 years who were not pregnant or lactating and had complete information for age and 24-hour dietary recall on day one. ^3^ Unlike the National Cancer Institute-adjusted statistics that represent long-term, usual dietary intake estimates presented in Table 3, the food intake statistics here are estimates of intake on any given day. The above sample sizes reflect a categorization of the NHANES respondents based on whether they reported sodium intakes in one of the above categories, and describe the sample, not the population. * Estimates have SEs of concern or sample sizes of concern.

**Table 7 nutrients-11-02691-t007:** Potential sodium intake reduction with MSG in the diets of U.S. children (aged 1–18 years) who consume selected food groups by age and select food group, NHANES 2013–2016 ^1,2,3^.

Food Group	*n*	Potential Reduction, Mean (SE)	Current Intake, Mean (SE)	Potential Intake After Reduction, Mean (SE)	Reduction (%)
Overall sodium (*Reduction among consumers of ≥1 food groups*)					
All consumers	3421	237 (0.7)	3047 (30.8)	2810 (30.1)	7.8 (0.02)
Boys	1717	263 (1.2)	3345 (46.3)	3082 (45.1)	8.5 (0.02)
Girls	1704	211 (2.9)	2747 (38.8)	2536 (35.9)	7.7 (0.1)
Select cheeses					
All consumers	1866	125 (3.4)	277 (7.6)	153 (4.2)	45.0 (0.0)
Boys	921	134 (5.2)	297 (11.5)	163 (6.3)	45.0 (0.0)
Girls	945	116 (3.7)	257 (8.1)	141 (4.5)	45.0 (0.0)
Cured meats					
All consumers	1166	255 (9.1)	636 (22.8)	382 (13.7)	40.0 (0.0)
Boys	620	271 (11.5)	677 (28.7)	406 (17.2)	40.0 (0.0)
Girls	546	235 (11.9)	588 (29.7)	353 (17.8)	40.0 (0.0)
Meat-based frozen meals					
All consumers *	14	196 (14.1)	783 (56.5)	587 (42.4)	25.0 (0.0)
Boys *	9	224 (20.6)	895 (82.5)	671 (61.9)	25.0 (0.0)
Girls *	5	153 (15.3)	613 (61.0)	460 (45.8)	25.0 (0.0)
Meat-based soups					
All consumers	175	329 (21.9)	1097 (73.1)	768 (51.2)	30.0 (0.0)
Boys	89	350 (37.8)	1168 (126.1)	818 (88.3)	30.0 (0.0)
Girls	86	307 (32.5)	1024 (108.3)	717 (75.8)	30.0 (0.0)
Meat-based gravies					
All consumers	107	122 (17.0)	405 (56.6)	284 (39.6)	30.0 (0.0)
Boys	62	144 (22.0)	479 (73.2)	336 (51.2)	30.0 (0.0)
Girls	45	87 (14.5)	292 (48.5)	204 (33.9)	30.0 (0.0)
Crackers and salty snacks					
All consumers	1618	109 (6.1)	273 (15.3)	164 (9.2)	40.0 (0.0)
Boys	814	111 (6.0)	277 (15.0)	166 (9.0)	40.0 (0.0)
Girls	804	107 (9.3)	268 (23.2)	161 (13.9)	40.0 (0.0)
Vegetable soups					
All consumers	111	221 (27.8)	737 (92.6)	516 (64.8)	30.0 (0.0)
Boys *	51	244 (43.6)	812 (145.2) *	569 (101.7) *	30.0 (0.0)
Girls	60	192 (18.7)	639 (62.5)	447 (43.7)	30.0 (0.0)

^1^ MSG, monosodium glutamate; NHANES, National Health and Nutrition Examination Survey; SE, standard error. ^2^ The analytic sample includes individuals 1–18 years old who were not pregnant or lactating and had complete information for age and 24-hour dietary recall on day one. ^3^ Unlike the National Cancer Institute-adjusted statistics that represent long-term, usual dietary intake estimates presented in Table 3, the food intake statistics here are estimates of intake on any given day. The above sample sizes reflect a categorization of the NHANES respondents based on whether they reported sodium intakes in one of the above categories, and describe the sample, not the population. * Estimates have SEs of concern or sample sizes of concern.

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
