# Peer review of "Current Sodium Intakes in the United States and the Modeled Effects of Glutamate Incorporation into Select Savory Products"

_nutrients, 2019, doi:10.3390/nu11112691_

Round 1

Reviewer 1 Report

This study described the estimates of contemporary sodium intakes of the U.S. population and then to model glutamate substitution in select products using NHANES database. The addition of glutamates to certain savory food categories has the potential to help reduce the population’s intake of sodium by approximately 3.0% and by 7.3% among consumers of the product categories in which glutamates were substituted for sodium chloride. This is concise and well-written but I have some comments for consideration.

When % contribution of sodium intake in all the savory products listed in Table 4 or 5 are summed, it is less than 12%. On the other hands, breads (of any kinds) contribute the most sodium to the U.S. diet. What is the % contribution of sodium intake from breads? If the contribution of breads is substantially large, it may be a better approach to reduce sodium contained in breads at least at the population level. Is it possible to incorporate glutamate substituted for sodium chloride in the breads without losing taste? Page 6, ‘For example, meat-based frozen meals provide 0.3% of the sodium present in the diets of all U.S adults, but as much as one-third (32%) among consumers.’ This example is meaningless at the population level because less than 1% consumes meat-based frozen meals when I see table 4. Too much abbreviations, such as NASEM, et.al. Lists of abbreviations may be helpful for the readers.

Author Response

Reviewer #1

This study described the estimates of contemporary sodium intakes of the U.S. population and then to model glutamate substitution in select products using NHANES database. The addition of glutamates to certain savory food categories has the potential to help reduce the population’s intake of sodium by approximately 3.0% and by 7.3% among consumers of the product categories in which glutamates were substituted for sodium chloride. This is concise and well-written, but I have some comments for consideration.

Author Response:Thank you for the very thoughtful and constructive comments.  We have addressed each one below.

Minor Comments:

When % contribution of sodium intake in all the savory products listed in Table 4 or 5 are summed, it is less than 12%. On the other hands, breads (of any kinds) contribute the most sodium to the U.S. diet. What is the % contribution of sodium intake from breads?

Author Response: Thank you for the comment.  According to NHANES 2011-2012, Breads, Rolls, and Tortillas contribute 7.9% of total sodium intake among all U.S. adults (≥20y) and 12.9% of total sodium intake among adult consumers (WWEIA, NHANES 2011-2012, Prepared by Food Surveys Research Group, Beltsville Human Nutrition Research Center, USDA).

If the contribution of breads is substantially large, it may be a better approach to reduce sodium contained in breads at least at the population level. Is it possible to incorporate glutamate substituted for sodium chloride in the breads without losing taste?

Author Response:   Incorporating glutamate as a substitute for sodium chloride in breads is not possible without losing taste. In order to do so, it would require incorporation of chemicals that would potentially impact the look and palatability of bread, reducing the likelihood of consumer purchase.

Page 6, ‘For example, meat-based frozen meals provide 0.3% of the sodium present in the diets of all U.S adults, but as much as one-third (32%) among consumers.’ This example is meaningless at the population level because less than 1% consumes meat-based frozen meals when I see table 4.

Author Response: This is an illustrious example of one extreme of sodium intake; it is designed to illustrate why we need to take into consideration the different patterns of intake of co-consumers.

Too much abbreviations, such as NASEM, et.al. Lists of abbreviations may be helpful for the readers.

Author Response: Nutrients author guidelines state “Abbreviations should be defined in parentheses the first time they appear in the abstract, main text and in figure or table captions and used consistently thereafter.”

Abbreviations we include are: DALYs, FDA, MSG, NHANES, NCHS, CDC, MEC, DRI, NASEM, CDRR, UL, USDA, NCI.

Reviewer 2 Report

The manuscript titled 'Current Sodium Intakes in the United States and the Modeled Effects of Glutamate Incorporation Into Select Savory Products' is a very interesting piece to read. The manuscript is well organized and written. The only comment I have is about statistics, in this section, what exactly is the model that fitted seems to be missing. Please specify it even if you have referred.

One note is about MSG, there might be a difference of acceptance between East and West. MSG was used commonly in cooking and savory products every where in the East, but nowadays, the view from the mass has changed to that MSG may not be 'healthy', since it is 'chemical' and 'not natural', synthesized in the lab. I wonder how consumers in the US perceive it.

Author Response

The manuscript titled 'Current Sodium Intakes in the United States and the Modeled Effects of Glutamate Incorporation into Select Savory Products' is a very interesting piece to read. The manuscript is well organized and written.

Author Response:Thank you for the very thoughtful and constructive comments.  We have addressed each one below.

Minor Comments:

The only comment I have is about statistics, in this section, what exactly is the model that fitted seems to be missing. Please specify it even if you have referred.

Author Response:In the section “SAS macros necessary to fit this model and to perform estimation of usual intake distributions, as well as additional details and resources concerning the NCI Method, are available via the NCI website.” The fitted model we are referring to is a two-part model that first uses logistic regression to estimate the probability of intake consumption for each consumer, and then secondly uses linear regression to estimate the actual consumption day amount of intake on a transformed scale, while taking into account within-person variation.   (https://epi.grants.cancer.gov/diet/usualintakes/method.html).This has been inserted in to the text.

One note is about MSG, there might be a difference of acceptance between East and West. MSG was used commonly in cooking and savory products everywhere in the East, but nowadays, the view from the mass has changed to that MSG may not be 'healthy', since it is 'chemical' and 'not natural', synthesized in the lab. I wonder how consumers in the US perceive it.

Author Response:Two different papers discuss consumer perceptions of MSG in the United States (see citations below).   We added these two references and a sentence into the discussion section for clarity:

Miyaki, T. , Retiveau‐Krogmann, A. , Byrnes, E. and Takehana, S. (2016), Umami Increases Consumer Acceptability, and Perception of Sensory and Emotional Benefits without Compromising Health Benefit Perception. Journal of Food Science, 81: S483-S493. doi:10.1111/1750-3841.13195

Wang, S, Adhikari, K. Consumer perceptions and other influencing factors about monosodium glutamate in the United States. J Sens Stud. 2018; 33:e12437. https://doi.org/10.1111/joss.12437